# Does living close to a petrochemical complex increase the adverse psychological effects of the COVID-19 lockdown?

**Paloma Vicens**[1,2], **Luis Heredia**[1,2], **Edgar Bustamante**[3], **Yolanda Pérez**[3], **José L. Domingo**[2], **Margarita Torrente**[1,2¤]*

**1** Department of Psychology, CRAMC (Research Center for Behavior Assessment), Universitat Rovira i Virgili, Tarragona, Spain, **2** Laboratory of Toxicology and Environmental Health, TECNATOX, School of Medicine, Universitat Rovira i Virgili, Reus, Spain, **3** Department of Geography, GRATET, Universitat Rovira i Virgili, Vila-seca, Spain

¤ Current address: Department of Psychology, Campus Sescelades, Universitat Rovira I Virgili, Tarragona, Spain

* margarita.torrente@urv.cat

**Data Availability Statement:** All relevant data are within the paper and also raw data is available from the DANS repository under the following

## Abstract

The petrochemical industry has made the economic development of many local communities possible, increasing employment opportunities and generating a complex network of closely-related secondary industries. However, it is known that petrochemical industries emit air pollutants, which have been related to different negative effects on mental health. In addition, many people around the world are being exposed to highly stressful situations deriving from the COVID-19 pandemic and the lockdowns adopted by national and regional governments. The present study aims to analyse the possible differential effects on various psychological outcomes (stress, anxiety, depression and emotional regulation strategies) stemming from the COVID-19 pandemic and consequent lockdown experienced by individuals living near an important petrochemical complex and subjects living in other areas, non-exposed to the characteristic environmental pollutants emitted by these kinds of complex. The sample consisted of 1607 subjects who answered an ad hoc questionnaire on lockdown conditions, the Perceived Stress Scale (PSS), the Hospital Anxiety and Depression Scale (HADS), the Barratt Impulsivity Scale (BIS) and the Emotional Regulation Questionnaire (ERQ). The results indicate that people living closer to petrochemical complexes reported greater risk perception [$K = 73.42$, $p < 0.001$, with a medium size effect ($\eta^2 = 0.061$)]. However, no significant relationship between psychological variables and proximity to the focus was detected when comparing people living near to or far away from a chemical/petrochemical complex. Regarding the adverse psychological effects of the first lockdown due to COVID-19 on the general population in Catalonia, we can conclude that the conditions included in this survey were mainly related to changes in the participants' impulsivity levels, with different total impulsivity scores being obtained if they had minors in their care ($p < 0.001$), if they had lost their jobs, if they were working ($p < 0.001$), if they were not telecommuting ($p < 0.001$), if they went out to work ($p < 0.001$) or if they established routines ($p = 0.009$). However, we can also be fairly certain that the economic effects are going to be

identification number: https://doi.org/10.17026/
dans-za8-qpmw.

**Funding:** The authors received no specific funding
for this work.

**Competing interests:** The authors have declared
that no competing interests exist.

worse than those initially detected in this study. More research will be necessary to corroborate our results.

## Introduction

Over the past 200 years the world has undergone a rapid and continuous industrialization process. Of the industrial sectors, petroleum-related activities have formed the core of this development. Today there are many petrochemical complexes all over the world. The petrochemical industry has brought about the economic development of many local communities, increasing employment opportunities and generating a complex network of closely-related secondary industries [1]. However, living near to these industrial complexes also brings concerns. It is known that petrochemical industries emit air pollutants that have been related to different negative effects on human health [2]. Studies have shown that compounds such as sulphur dioxide ($SO_2$), particulate matter (PM), polycyclic aromatic hydrocarbons (PAHs) and volatile organic compounds (VOCs) are frequently found in the ambient air around petrochemical complexes [3–5]. Meanwhile other studies have related human exposure to these compounds with an increase in cancer mortality, acute lower respiratory infections, asthma and cardiovascular diseases [6–8]. In addition, living in areas close to petrochemical complexes has also been related to increases in the occurrence of hypothyroidism [9] and pre-term births [10].

The mental health of people living near petrochemical complexes can be also affected because some of the pollutants (e.g. PM and $NO_2$) emitted by these facilities are linked to oxidative stress and inflammatory processes in the brain [11, 12]. Zhang, Wang [13] assessed the neuropsychological function of a group of petrochemical workers and a group of office personnel from the same facility, observing decreased learning and working memory in petrochemical workers compared to the office personnel group. This decreased working memory function has also been seen in children, depending on the distance between their normal place of residence and the site of the petrochemical complex [14]. However, the working memory function is not the only neuropsychological function that can be affected by the presence of air pollutants in the area surrounding petrochemical complexes. In a community-based previous study by Vichit-Vadakan, Vajanapoom [15], a sample of 17,515 participants living within a 10-km radius of petrochemical industries were assessed for neuropsychological performance. The results showed that those living near these complexes performed worse on tests that assessed eye-hand coordination, short-term recall, and hand and eye movement responsiveness. Moreover, participants living less than 3 km from the centre of the industrial complex were more likely to exhibit forgetfulness, anxiety, depression and loss of concentration. Finally, it is important also to consider the possible impact on the population's mental health stemming from the occurrence of accidents in nearby petrochemical complexes. A community-based study by Peek, Cutchin [16] reported decreases in local residents' self-perceived mental and physical health after experiencing an incident at the plant, even in those who were not directly affected by it.

One variable that could be related to the effects mentioned above is subjective risk perception. Some authors have argued that one's subjective risk perception of environmental threats leads to chronic stress because of the fear of potential health problems, the uncertainty of the threat and the lack of control over it [17]. However, previous studies have reported incongruent results regarding the stress levels of people living near petrochemical industries. Thus

while some researchers have reported increased stress levels in exposed areas [18, 19], others have observed no significant differences [20]. Axelsson, Stockfelt [21] assessed anger and worry in individuals living in the vicinity of this kind of complex (< 3 km–exposed group) and subjects residing in a control area (> 24 km–nonexposed group). The results showed that there were more than twice as many subjects who frequently worried about the health effects of industrial air pollution in the exposed group than in the nonexposed group. Similarly, the number of subjects worried about accidents in industrial activities was three times higher among people living in the exposed area than in the nonexposed area.

Bearing in mind that living around industrial facilities could be a health risk factor, this population would be considered vulnerable to unexpected stressful situations since they have higher basal levels of stress and worry [21].

Nowadays many people around the world are being exposed to highly stressful situations deriving from the COVID-19 pandemic and the lockdowns imposed by national and regional governments. Brooks, Webster [22] recently reviewed studies that focused on the mental health consequences of previous severe acute respiratory syndrome (SARS), H1N1 influenza and Ebola outbreaks. They concluded that the most frequent psychological outcomes in quarantined people were acute stress disorders, trauma-related mental health disorders and depressive symptoms, with the most frequently reported emotional states being fear, nervousness, sadness and guilt. The authors also concluded that being a woman, working in the health services, losing economic income and being quarantined for more than 10 days were factors associated with poorer mental health [22]. For this reason the scientific community is making a great effort to understand the psychological consequences of the COVID-19 pandemic. In a recent study by Odriozola-González, Planchuelo-Gómez [23], a group of 2530 university members were examined for anxiety, depression and stress by way of an online survey during the high lockdown period in Spain. The percentage of responses scoring in the moderate to extremely range for several of the variables assessed was 21.34% for anxiety, 34.19% for depression and 28.14% for stress. Using a different methodology, Li, Wang [24] assessed a sample of 17,865 Weibo (a Chinese microblogging network) users via a learning machine algorithm focusing on the number of positive and negative words found in their posts. The results showed an increase in the number of words related to negative emotions (anxiety, depression and indignation), while references to positive emotions decreased. It has been suggested that in stressful situations a subject's ability to manage their emotions could be a crucial way of mitigating the downstream of negative consequences in the lockdown period [25]. These studies could be pointing to a serious effect on the population's depressive mood due to the lockdown procedures during the COVID-19 outbreak. In this regard, a recent review of community-based studies on the prevalence of depression during 2020 has reported a prevalence of 25%, a percentage seven times higher than the global estimate for depression in 2017 (3.44%) [26]. However, there is currently a dearth of knowledge about the effects of lockdown on the population's mental health during the COVID-19 outbreak.

Taking all the above into account, the aim of the present study is to analyse the possible differential effects on various psychological outcomes (stress, anxiety, depression and emotional regulation strategies) deriving from the COVID-19 pandemic and consequent lockdown experienced by individuals living near a large petrochemical complex and subjects living in other areas, non-exposed to the typical environmental pollutants emitted by these kinds of complex. The main objective is to study whether people living near a petrochemical complex showed a greater psychological impact during the lockdown than those living in non-exposed areas. To this end, the following specific objectives were established:

1. To explore any relationship to COVID risk perception and lockdown variables.

2. To assess the risk perception related to living in the vicinity of a chemical/petrochemical complex.

3. To evaluate the relationship between the distance to a chemical/petrochemical complex and psychological variables such as perceived stress, impulsivity, anxiety and depression symptomatology, and emotional regulation.

4. To analyse whether psychological variables were affected by lockdown conditions.

5. To assess which variables (closeness to petrochemical/chemical complex, COVID-19 and lockdown conditions) influence the psychological state.

The present study wanted to test the following hypotheses:

1. During the first COVID lockdown a high percentage of people would think that COVID-19 was dangerous.

2. People closer to a petrochemical complex would be more worried about the danger of these complexes.

3. Subjects living near the complex would show higher levels of psychological impact during lockdown compared to individuals living far away from the area of environmental pollution.

4. COVID lockdown conditions would make subjects score differentially in the psychological outcomes.

5. The fact of living close to a petrochemical complex and COVID lockdown conditions affect the psychological outcomes studied.

## Materials and methods

### Area of study

The province of Tarragona (Catalonia, Spain) is home to a sizeable concentration of industrial activity. Since the 1960s an increasing number of chemical and petrochemical companies–including a big oil refinery, a chlor-alkali plant and various plastic manufacture chemical companies along with a municipal solid waste incinerator and a hazardous waste incinerator–have become established in the area, turning it into the most important petrochemical complex in Southern Europe.

### Experimental design

The study consisted of an *ex post facto* correlational and comparative design. The variables assessed were the place of habitual residence and its proximity to the chemical/petrochemical complex, risk perception regarding the closeness of the petrochemical complex and the spread of the COVID-19 pandemic, lockdown conditions, self-reported levels of stress, anxiety, depression, impulsivity, and cognitive emotional regulation strategies used during lockdown. On the basis of data in the previous literature on the adverse health effects associated with living near chemical/petrochemical complexes, the participants were divided into two groups: those living near ($\leq 10$ km) or far from ($> 10$ km) the petrochemical complex [6, 27, 28]. An online survey (*Petrocovid Survey* -PS- see Tables 1 and 2) was used to obtain the data. Voluntary participants were able to complete the survey from April 22 to May 14 2020. The evolution of the number of COVID-19 cases in Spain, the different levels of lockdown applied in the course of the health emergency and the assessment period of the present study are shown in

**Table 1. Items included in the first section of the Petrocovid Survey (PS).**

| Items | Response options |
|---|---|
| 1. Age: | In years |
| 2. Gender: | Male / Female |
| 3. Place of residence during lockdown: | Town or city name |
| Postcode: | |
| 4. Place of habitual residence: | Town or city name |
| Postcode: | |
| 5. Is your habitual residence near a chemical or petrochemical complex? | Yes / No |
| 6. If so, have you perceived the location of your residence as dangerous? | Low ('slightly or nothing')/Medium ('quite/fairly')/ High ('a lot') |

Fig 1. It can be seen that the assessment period partially coincided with the initial lifting of restrictions. The data were stored anonymously on the computer server of the Psychology Department at the Universitat Rovira i Virgili (Tarragona, Catalonia, Spain).

To determine the distance between each participant's habitual residence and the emission sources of environmental pollutants from the petrochemical complex, the province of Tarragona and its neighbouring provinces (Castelló, Teruel, Zaragoza, Lleida and Barcelona, Fig 2) were taken as a spatial reference. In other words, only surveys from these provinces were taken into account.

The cartography was created using ArcGIS 10.2.3 software (Environmental Systems Research Institute, Redlands, CA, USA) in a point vector format using ETRS89 UTM 31N projection.

The postcode layer was generated using the municipal polygonal base from the Centro Nacional de Información Geográfica (CNIG). In order to obtain the postcode point layer, first

**Table 2. Items included in the second section of the Petrocovid Survey (PS).**

| Items | Response options |
|---|---|
| 1. Have you been alone at home? | Yes / No |
| 2. If accompanied, were you with: | |
| Minor(s) in your care | Yes / No |
| People over age 65 | Yes / No |
| Dependent(s) in your care | Yes / No |
| 3. Have you had COVID-19? | Yes / No |
| 4. Are you part of the at-risk population? | Yes / No |
| 5. Have any of your family members or friends died? | Yes / No |
| 6. Have you lost your job during lockdown? | Yes / No |
| 7. Has your company requested an ERTE (Spanish version of the furlough scheme)? | Yes / No |
| 8. Have you worked during lockdown? | Yes / No |
| 9. Have you been able to telecommute? | Yes / No |
| 10. Have you left home to go to work? | Yes / No |
| If so, in which field do you work? | Health service / Food industry / Petrochemical / Other |
| 11. Regardless of whether you have gone out to work or not, how often do you leave home in a week? | Once a week / 2–4 times a week / More than 4 times a week |
| 12. Have you established routines during lockdown? | Yes / No |
| 13. Have you perceived the situation arising from COVID-19 as dangerous? | Low ('slightly or nothinng')/Medium ('quite/ fairly')/High ('a lot') |

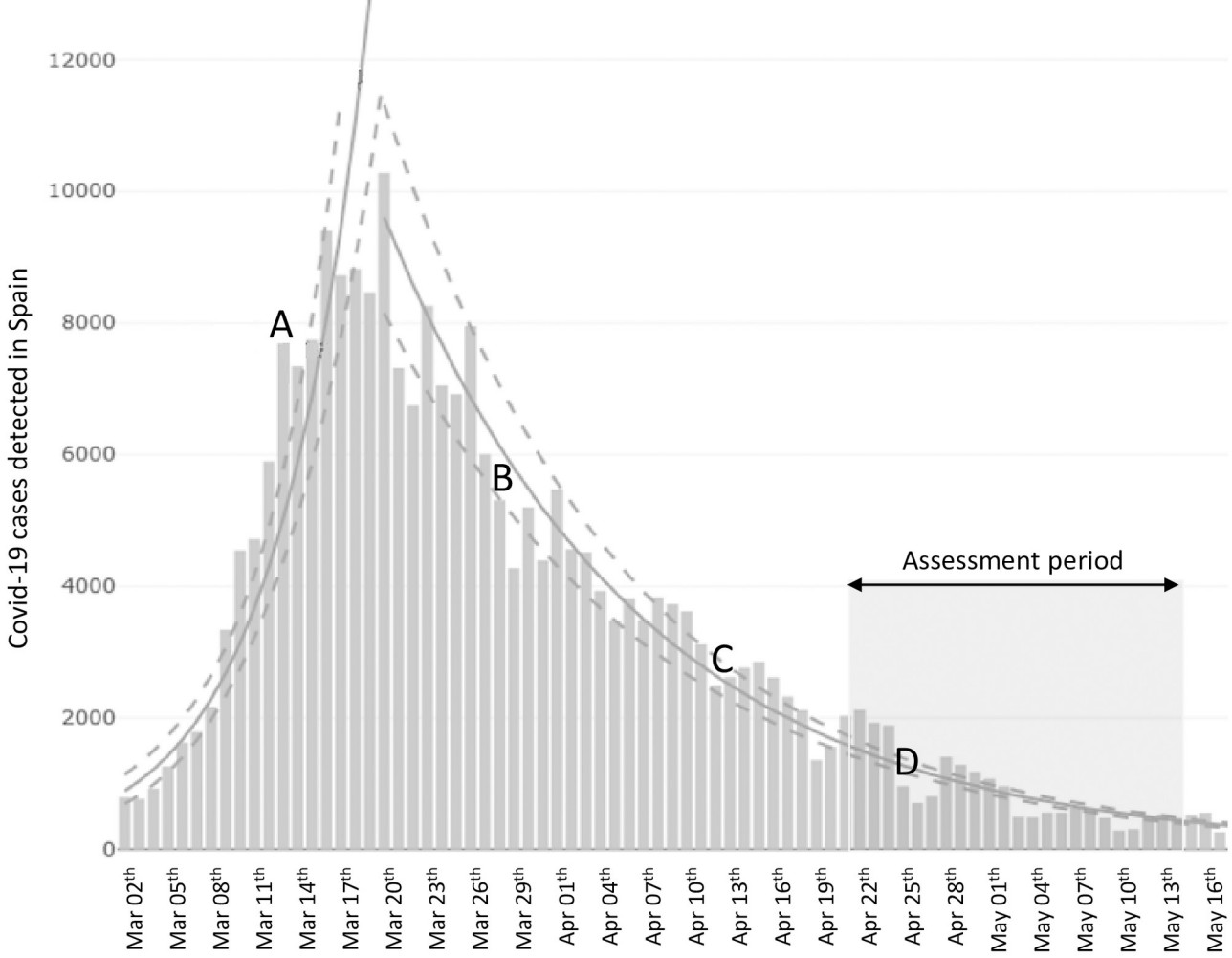

A: Spanish government announce the national lockdown (medium lockdown).
B: Spanish government announce the lockdown of workers of all non essential services (high lockdown).
C: Some non essential workers can go back to the work (medium lockdown).
D: Start of the lockdown de-escalation.
Source: National Epidemiological Surveillance Network (RENAVE). Carlos III Health Institute (ISCIII). Spain.

**Fig 1. Total daily number of COVID-19 cases detected in Spain during lockdown and the Petrocovid Survey (PS) assessment period.**

of all the borders between adjacent municipalities with the same postcode were removed, thereby providing a single polygon for each postcode area. Secondly, the midpoint of each polygon of that layer was generated. The layer showing pollutant emission sources from petro-chemical industries was created by digitizing the points appearing in the Industrial Estates Database belonging to the Institut Cartogràfic i Geològic de Catalunya (ICGC).

In studies of this type it is normal to use the Euclidean distance to establish the relationship between the focus of contamination and the reference locations (from habitual homes or schools to major roadways, foundries, mineral storage areas or airports, for example) [29–34]. Once the two point-type layers were obtained, the straight-line distance in metres between the postcode (place of residence) and the nearest source of pollution was calculated using the

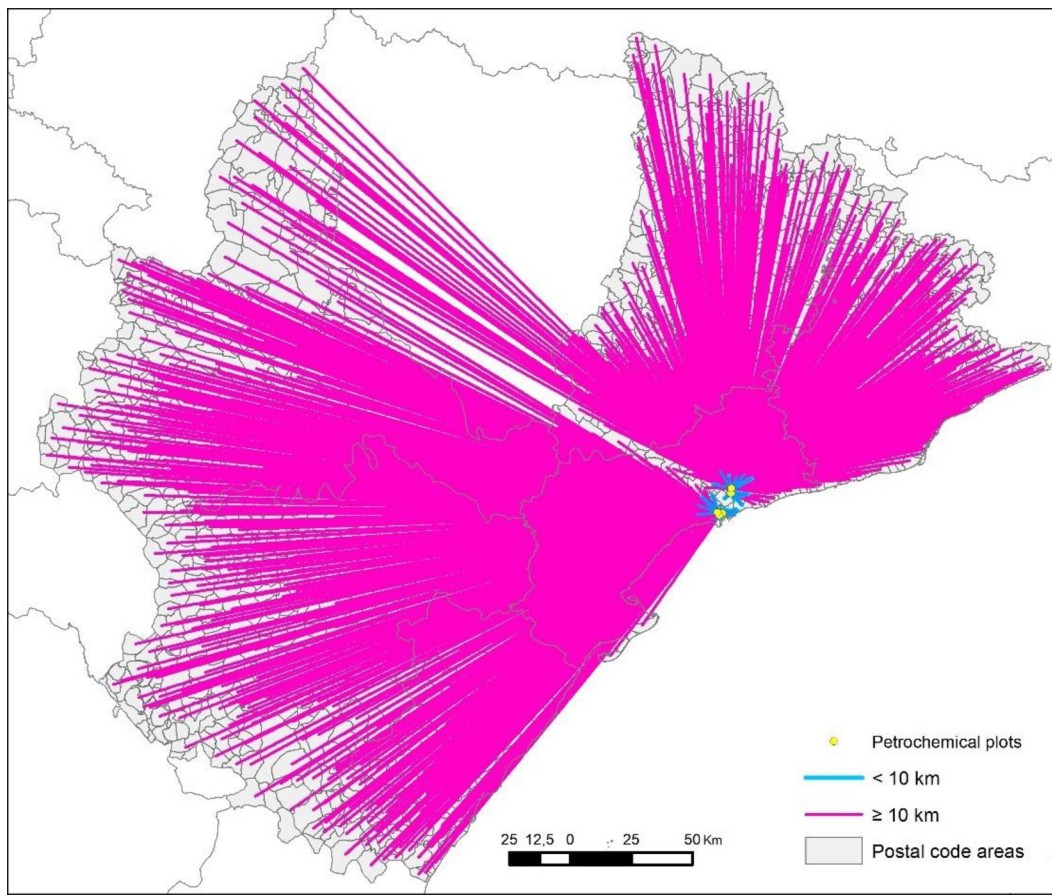

**Fig 2. Distance between the place of residence (survey locations) and the nearest source of petrochemical pollutants.**
Map based on BDLJE CC-BY scne.es.

'nearest' option. Fig 2 shows the 1607 survey locations and the straight-line distance to the nearest source of contamination.

## Participants

The snowball sampling strategy was used to recruit participants by disseminating a computerized version of the Petrocovid Survey (PS) in social networks and the digital press. The survey was answered by 1655 subjects. The data were reviewed and five participants eliminated because they had not completed some of the PS questions. In the end the sample comprised 1607 subjects (1195 women, average age 39.43 ± 13.61 years, and 412 men, average age 44.02 ± 14.42 years). Of these, 1097 lived within 10 km of the petrochemical complex, while 510 lived over 10 km away. Before participants completed the survey, they were given information about the objectives of the study and an informed consent was requested. If the participants were minors or did not accept the ethics consent on the online platform, the survey ended without any data being collected. All the data obtained came from participants over the age of 18 who consented to their data being collected by the researchers. The present study followed the ethical principles of the World Medical Association's Declaration of Helsinki (revised in Tokyo in 2004). It was approved by the Ethics Committee of Clinical Research of the Pere Virgili Health Research Institute (IISPV, Tarragona, Catalonia, Spain) (reference

number: 084/2020). All data were processed in accordance with the European General Data Protection Regulation (GDPR) (2016/679).

## Instruments

**Petrocovid Survey (PS).** To determine the habitual place of residence and lockdown conditions, an online *ad hoc* survey was created. In the first section (Table 1) the subject was asked their age, gender, place of residence during lockdown and habitual residence, in both these cases providing the postcode. Subjects were also asked about the proximity of their habitual residence to the petrochemical complex (Yes/No) and their subjective risk perception on this location (Low/Medium/High).

The second section of the survey included data about lockdown conditions. These are summarized in Table 2.

**Perceived Stress Scale (PSS).** This was designed to measure the degree to which individuals consider situations in their lives to be stressful [35]. It consists of 14 items rated on a 5-point Likert scale ranging from 1 (never) to 5 (very often). The PSS was adapted for the Spanish population by Remor [36] with a Cronbach's alpha of .82 and a test-retest reliability coefficient of .73

**Hospital Anxiety and Depression Scale (HADS).** The HADS was originally developed to assess emotional distress (anxiety and depression) in samples of people suffering from physical illness [37]. However, it has also been used in the healthy population [38]. It is composed of 14 items with four response options (seven items for the anxiety subscale (HADA) and seven for the depression subscale (HADD)). It was validated for the Spanish population by Terol, López-Roig [39] with a Cronbach's alpha of .77 for anxiety and .71 for the depression subscale. The test-retest reliability coefficient was .77 for HADA and .74 for HADD.

**Barratt Impulsivity Scale (BIS).** This measures three traits of impulsiveness (motor, cognitive and non-planning impulsivity) [40]. It contains a total of 30 items, providing a total level of impulsiveness through a 4-point Likert scale ranging from 1 (never/almost never) to 4 (always/almost always). It was validated for the Spanish population obtaining a Cronbach's alpha of 0.67 for motor impulsivity and 0.59 for both cognitive impulsivity and non-planning impulsivity [41].

**Emotional Regulation Questionnaire (ERQ).** This is composed of 10 items, 6 assessing cognitive reappraisal and 4 assessing expressive suppression using a 7-point Likert scale (1 = strongly disagree; 7 = strongly agree) [42]. The items include questions about the regulation of both positive and negative emotions. It was validated for the Spanish population by Cabello, Salguero [43] with a Cronbach's alpha and test-retest reliability of 0.75 and 0.66 respectively for suppression and 0.79 and 0.64 for reappraisal.

## Statistical analysis

Group proportions were calculated for the categorical variables. To test for association we performed chi-squared or Fisher exact tests. The effect size was evaluated with the eta squared ($\eta^2$) based on the H-statistic coefficient [44]. For the continuous variables, a Shapiro-Wilk test was performed to assess normality. If the distribution of the scores was normal, we reported averages and standard deviations and performed t-tests. Otherwise we reported the median and interquartile range, employing Mann-Whitney U tests. The effect size was evaluated using Cohen's d.

For the multivariable analysis, we carried out linear regressions for the continuous variables and logistic regressions for the binomial. Mode selection was performed using the Akaike information criterion (AIC) and an automatic stepwise strategy, with forward and backward

steps. The model with the lowest AIC was automatically selected. Probability levels below 0.05 ($p < 0.05$) were considered statistically significant. The data were analysed using the R statistical software package, version 4.0.

# Results

## Hypothesis 1

For the question "Have you perceived the situation arising from COVID-19 as dangerous?", we obtained a frequency of 519 answering 'a lot' (high), 881 answering 'quite/fairly' (medium), and 207 answering 'slightly or nothing' (low), out of the total sample of 1607.

In order to explore any potential relationship between COVID-19 risk perception and the lockdown variables, the chi-squared test was used. Statistical analysis showed no significant relationships between the risk perception of COVID-19 and any other variables included in the second section of the Petrocovid Survey (see Table 2).

## Hypothesis 2

It is important to note that a significant relationship was observed when the risk perception associated with proximity to a chemical/petrochemical complex was assessed, with those individuals living closest to the chemical/petrochemical industries being the most worried. A logarithm's continuous variable of distance from home to the focus was used to calculate the analyses [$K = 73.42$, $p < 0.001$, with a medium size effect ($\eta^2 = 0.061$)].

## Hypothesis 3

In order to assess the relationship between the distance to the chemical/petrochemical complex and the psychological variables during lockdown, the fact of being at a distance of over 10 km from the chemical/petrochemical complex was considered as a criterion. Thus our sample was divided into two experimental groups: near (n = 583) and far (n = 1024). In this regard it is interesting to note that no differences in perceived stress, impulsivity, anxiety symptoms, depressive symptoms or emotional regulation were observed between the experimental groups.

## Hypothesis 4

Considering the relationship between the variables relating to the COVID-19 lockdown and the psychological variables, the results showed that **perceived stress** is affected by losing one's job [$W = 95371$, $p < 0.001$; *Cohen's d* = 0.51 (medium)], while people who continued working had less perceived stress [$W = 360551$, $p < 0.001$; *Cohen's d* = 0.37 (small)]. Interestingly, just going out influenced perceived stress [$K = 10.125$; $p = 0.006$; *Cohen's d* = 0.20 (small)]. A Tukey post-hoc test showed that people who answered they went out 'quite' had less perceived stress than subjects who responded 'slightly or nothing' [$p = 0.004$, *Cohen's d* = 0.21 (small)] (see Table 3).

As far as **impulsivity** measurements are concerned, subjects having minors in their care had a lower total impulsivity score [($W = 290120$, $p < 0.001$)] with a *Cohen's d* of 0.27 (small). The results showed that these individuals also had lower motor [$W = 279422$, $p < 0.001$; *Cohen's d* = 0.20 (small)] and cognitive [$W = 304254$, $p < 0.001$; *Cohen's d* = 0.36 (small)] impulsivity. People who had lost their jobs had a higher total impulsivity score [$W = 86807$, $p < 0.001$ *Cohen's d* = 0.35 (small)], as well as higher motor [$W = 89027$, $p < 0.001$ *Cohen's d* = -0.35

**Table 3. Variables that were significant as regards perceived stress.**

| Variable | Group | Mean | SD | SEM | n | p values |
|----------|-------|------|-----|-----|-----|----------|
| **Losing Job** | no | 22.64 | 9.16 | 0.24 | 1423 | p< 0.001 |
| | yes | 27,39 | 9.86 | 0.73 | 184 | |
| **Working** | no | 25.37 | 10 | 0.41 | 599 | p< 0.001 |
| | yes | 21.88 | 8.71 | | 1008 | |
| **Going out** | slightly or nothing | 23.6 | 9.52 | 0.29 | 1084 | p = 0.006 |
| | Quite | 21.61 | 8.67 | 0.5 | 300 | |
| | A lot | 23.28 | 9.31 | 0.62 | 223 | |

SD: Standard deviation, SEM: Standard error of mean.

(small)] and cognitive impulsivity [$W$ = 84418, p < 0.001 *Cohen's d* = -0.38 (small)]. Another significant result was observed for *people not working*. This group of population showed higher total impulsivity [$W$ = 309688, p < 0.001 *Cohen's d* = 0.47 (small)] together with higher motor [$W$ = 300880, p < 0.001; *Cohen's d* = 0.40 (small)] and cognitive impulsivity [$W$ = 300269, p < 0.001; *Cohen's d* = 0.40 (small)] as well as a greater lack of planning [$W$ = 294811, p < 0.001; *Cohen's d* = 0.35 (small)]. Similar results were found for those *not telecommuting*, who showed a greater lack of planning [$W$ = 304419, p < 0.001; *Cohen's d* = 0.28 (small)] along with higher total impulsivity [$W$ = 307133, p < 0.001; *Cohen's d* = 0.31 (small)] and motor impulsivity [$W$ = 305254, p < 0.001; *Cohen's d* = 0.30 (small)] scores. In turn, *establishing routines* was associated with a lower total impulsivity score [K = 6.3493, p = 0.011; *Cohen's d* = 0.33 (small)] together with a smaller lack of planning [K = 4.1318, p = 0.041; *Cohen's d* = 0.25 (small)] and lower motor impulsivity [K = 7.4641, p = 0.006; *Cohen's d* = 0.33 (small)]. People *going out to work* showed lower total impulsivity [$W$ = 197106, p < 0.001; *Cohen's d* = 0.20 (small)] and lower cognitive impulsivity [$W$ = 204489, p < 0.001; *Cohen's d* = 0.27 (small)]. The results are summarized in Table 4.

It is important to note that *working in different sectors* has different effects on motor impulsivity (K = 9.2715; p = 0.009). The Tukey test showed that those working in the health sector had lower motor impulsivity than those working in the food (*p* = 0.015) or petrochemical (*p* = 0.044) sectors, with a $\eta^2$ = 0.00187 (small) (see Fig 3).

As regards **anxious and depressive symptomatology**, we observed that *having minors in their care* increased anxious symptomatology [$W$ = 24534s8, p < 0.001; *Cohen's d* = 0.32 (small)], while *having dependents* in their charge decreased it [$W$ = 95597, p = 0.029; *Cohen's d* = 0.20 (small)]. Moreover, *losing one's job* decreased these symptoms $W$ = 146640, p < 0.001; *Cohen's d* = 0.39 (small). However, *establishing routines* increased depressive symptoms [$W$ = 8886, *p* = 0.021; *Cohen's d* = 32 (small)] (see Table 5).

Finally, with regard to **emotional regulation**, living with *people over 65* [$W$ = 85283, p < 0.001; *Cohen's d* = -0.24 (small)] and living with *dependent or disabled people* [$W$ = 54803, p = 0.043; *Cohen's d* = -0.23 (small)] increased suppression strategy. However, *establishing routines* increased cognitive re-evaluations as an emotional strategy [$W$ = 6674, p = 0.017; *Cohen's d* = -0.21 (small)] (see Table 6).

## Hypothesis 5

In order to assess which variables (proximity to the petrochemical complex, COVID-19 and lockdown conditions) are influencing the psychological state, regression analyses were conducted for each psychological outcome score, in which all pertinent variables were included. The variance inflation factors (VIF) of all the variables were calculated to evaluate collinearity

**Table 4.  (a) COVID lockdown variables that were significantly related to impulsivity. (b) COVID lockdown variables that were significantly related to impulsivity.**

| Variable | | Group | Mean | SD | SEM | n | p values |
|---|---|---|---|---|---|---|---|
| **a** | | | | | | | |
| **Having minors in their care** | Motor impulsivity | No | 18.68 | 3.9 | 0.13 | 904 | p< 0.001 |
| | | Yes | 17.89 | 3.58 | 0.15 | 552 | |
| | Cognitive impulsivity | No | 16.3 | 3.58 | 0.12 | 904 | p< 0.001 |
| | | Yes | 15.04 | 3.33 | 0.14 | 552 | |
| | Total impulsivity | No | 59.48 | 10.03 | 0.33 | 904 | p< 0.001 |
| | | Yes | 56.81 | 9.5 | 0.4 | 552 | |
| **Losing one's job** | Motor impulsivity | No | 18.23 | 3.66 | 0.1 | 1294 | p< 0.001 |
| | | Yes | 19.56 | 4.64 | 0.36 | 162 | |
| | Cognitive impulsivity | No | 15.67 | 3.44 | 0.1 | 1294 | p< 0.001 |
| | | Yes | 17.02 | 4.06 | 0.32 | 162 | |
| | Total impulsivity | No | 58.07 | 9.65 | 0.27 | 1294 | p< 0.001 |
| | | Yes | 61.61 | 11.37 | 0.89 | 162 | |
| **Working** | Lack of planning | No | 25.32 | 4.74 | 0.21 | 533 | p< 0.001 |
| | | Yes | 23.65 | 4.66 | 0.15 | 923 | |
| | Motor impulsivity | No | 19.33 | 4.06 | 0.18 | 533 | p< 0.001 |
| | | Yes | 17.83 | 3.53 | 0.12 | 923 | |
| | Cognitive impulsivity | No | 16.72 | 3.72 | 0.16 | 533 | p< 0.001 |
| | | Yes | 15.3 | 3.32 | 0.11 | 923 | |
| | Total impulsivity | No | 61.38 | 10.12 | 0.44 | 533 | p< 0.001 |
| | | Yes | 56.78 | 9.4 | 0.31 | 923 | |
| **b** | | | | | | | |
| **Telecommuting** | Lack of planning | No | 25 | 4.88 | 0.19 | 645 | p< 0.001 |
| | | Yes | 23.68 | 4.58 | 0.16 | 811 | |
| | Motor impulsivity | No | 19.02 | 4.1 | 0.16 | 645 | p< 0.001 |
| | | Yes | 17.87 | 3.46 | 0.12 | 811 | |
| | Total impulsivity | No | 60.19 | 10.55 | 0.42 | 645 | p< 0.001 |
| | | Yes | 57.1 | 9.16 | 0.32 | 811 | |
| **Going out to work** | Cognitive impulsivity | No | 16.03 | 3.52 | 0.1 | 1151 | p< 0.001 |
| | | Yes | 15.05 | 3.49 | 0.2 | 305 | |
| | Total impulsivity | No | 58.88 | 9.66 | 0.28 | 1151 | p< 0.001 |
| | | Yes | 56.89 | 10.71 | 0.61 | 305 | |
| **Establishing routines** | Lack of planning | No | 24.94 | 4.47 | 0.56 | 63 | p = 0.041 |
| | | Yes | 23.64 | 5.2 | 0.3 | 292 | |
| | Motor impulsivity | No | 19.19 | 3.79 | 0.48 | 63 | p = 0.006 |
| | | Yes | 17.85 | 4.06 | 0.24 | 292 | |
| | Total impulsivity | No | 60.06 | 10.03 | 1.26 | 63 | p = 0.009 |
| | | Yes | 56.52 | 10.55 | 0.62 | 292 | |

SD: Standard deviation, SEM: Standard error of mean.

problems. All the VIF indexes showed values under 5, indicating an absence of collinearity between the variables included in each model [45].

**Perceived Stress Scale (PSS).**   The PSS total score is influenced by age ($\beta$ = -0.36, $p<$0.001), gender (being male) ($\beta$ = -0.18, $p$ = 0.001), and not responding regarding their perception of petrochemical complex danger ($\beta$ = -0.13, $p$ = 0.039). The PSS total score is also influenced by not living with people over age 65 ($\beta$ = 0.19, $p$ = 0.006), losing one's job ($\beta$ =

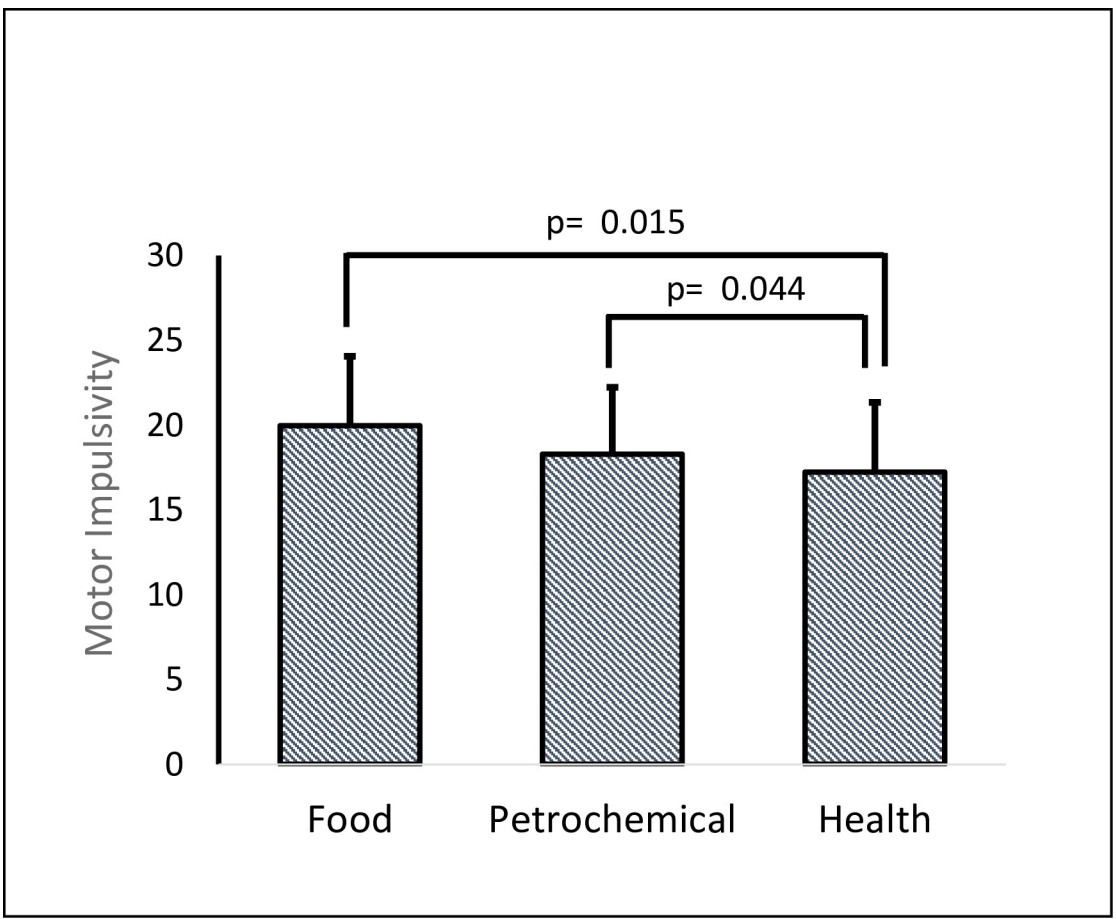

**Fig 3. Differences in motor impulsivity levels depending on employment sector.**

0.28, $p<0.001$), working during lockdown ($β = -0.24$, $p = 0.002$), perceiving the COVID-19 situation as very dangerous ($β = 0.23$, $p<0.001$) and perceiving it as nothing or slightly dangerous ($β = -0.42$, $p<0.001$). All the variables entered in the regression analysis (significant and non-significant) explain 22% of the PSS variance ($r^2$ adjusted = 0.22) (see Table 7).

**Barratt Impulsivity Scale (BIS).** The non-planning score of the BIS is influenced by gender (being male) ($β$: -0.12, $p = 0.040$), working during lockdown ($β = -0.30$, $p<0.001$),

**Table 5. Lockdown variables that were significantly related to anxious and depressive symptomatology.**

| Symptomatology | Variable | Group | Mean | SD | SEM | n | p values |
|---|---|---|---|---|---|---|---|
| **Anxious** | Minors in their care | No | 11.06 | 2.52 | 0.18 | 191 | p< 0.001 |
| | | Yes | 11.83 | 2.28 | 0.17 | 186 | |
| | Dependents in their charge | No | 11.46 | 2.4 | 0.06 | 1442 | p = 0.029 |
| | | Yes | 10.97 | 2.43 | 0.22 | 119 | |
| | Losing one's job | No | 11.53 | 2.34 | 0.06 | 1384 | p< 0.001 |
| | | Yes | 10.58 | 2.73 | 0.2 | 177 | |
| **Depressive** | Routines | No | 8.96 | 2.42 | 0.29 | 70 | p = 0.021 |
| | | Yes | 9.56 | 1.72 | 0.1 | 307 | |

SD: Standard deviation, SEM: Standard error of mean.

**Table 6. Lockdown variables that were significantly related to emotional regulation.**

| Strategy | Variable | Group | Mean | SD | SEM | n | p values |
|---|---|---|---|---|---|---|---|
| **Suppression** | People over 65 | No | 14.83 | 5.7 | 0.33 | 294 | p< 0.001 |
| | | Yes | 16.23 | 5.23 | 0.88 | 35 | |
| | | | | | | | |
| | Dependents in their care | No | 14.18 | 5.72 | 0.16 | 1247 | p = 0.043 |
| | | Yes | 15.5 | 5.7 | 0.57 | 100 | |
| **Re-evaluation** | Routines | No | 28.47 | 6.16 | 0.38 | 263 | p = 0.017 |
| | | Yes | 29.86 | 6.74 | 0.2 | 1084 | |

SD: Standard deviation, SEM: Standard error of mean.

establishing routines ($\beta$ = 0.32, $p$<0.001) and perceiving the COVID-19 situation as not or slightly dangerous ($\beta$ = -0.16, $p$ = 0.046). These variables (significant and non-significant) explain only around 5% of the non-planning score variance ($r^2$ adjusted = 0.049) (see Table 8).

The BIS motor score is influenced by living with children ($\beta$ = -0.14, $p$ = 0.011), working during lockdown ($\beta$ = -0.29, $p$<0.001), working in the health sector ($\beta$ = -0.39, $p$ = 0.019) and establishing routines during lockdown ($\beta$ = -0.36, $p$<0.001). All these variables (significant and non-significant) explain 7% of the motor score variance of the BIS ($r^2$ adjusted = 0.07) (see Table 9).

The cognitive score is influenced by age ($\beta$ = -0.24, $p$<0.001), gender (being male) ($\beta$ = -0.19, $p$ = 0.001), living with children ($\beta$ = -0.24, <0.001), working ($\beta$ = -0.27, $p$<0.001), telecommuting ($\beta$ = 0.13, $p$ = 0.047) and establishing routines during lockdown ($\beta$ = -0.23, $p$<0.001). These variables (significant and non-significant) explain 14% of the cognitive score variance ($r^2$ adjusted = 0.14) (see Table 10).

**Table 7. Regression analysis relating to perceived stress.**

| | TOTAL PSS | | |
|---|---|---|---|
| **Predictors** | **β** | **CI** | **p values** |
| **(Intercept[1])** | 0.06 | -0.06 – 0.17 | 0.321 |
| **Age** | -0.36 | -0.40 – (-0.31) | **<0.001** |
| **Gender [male]** | -0.18 | -0.28 – (-0.08) | 0.001 |
| **Distance from focus** | 0.04 | -0.01 – 0.09 | 0.105 |
| **Danger perception [slightly or nothing]** | -0.07 | -0.19 – 0.04 | 0.205 |
| **Danger perception [a lot]** | 0.11 | -0.05 – 0.26 | 0.170 |
| **Danger perception [no response]** | -0.13 | -0.26 – (-0.01) | **0.039** |
| **People over 65** | 0.19 | 0.05 – 0.32 | **0.006** |
| **Dependent or disabled people** | 0.15 | -0.02 – 0.32 | 0.077 |
| **Losing one's job** | 0.28 | 0.13 – 0.42 | **<0.001** |
| **Working** | -0.24 | -0.39 – (-0.09) | 0.002 |
| **Telecommuting** | 0.13 | -0.01 – 0.26 | 0.063 |
| **Going out to work** | 0.12 | -0.01 – 0.26 | 0.070 |
| **COVID-19 as dangerous [a lot]** | 0.23 | 0.13 – 0.33 | **<0.001** |
| **COVID-19 as dangerous [slightly or nothing]** | -0.42 | -0.56 – (-0.29) | **<0.001** |
| **Observations:** 1607 | | | |
| **R$^2$ / R$^2$ adjusted:** 0.228 / 0.221 | | | |

CI: Confidence interval.

[1]The **intercept** (or constant) is the expected mean value of Y when all X = 0.

**Table 8. Regression analysis relating to lack of planning in the BIS.**

| Predictors | LACK OF PLANNING | | |
|---|---|---|---|
| | β | CI | p values |
| (Intercept[1]) | 0.54 | 0.40 – 0.68 | **<0.001** |
| Gender [male] | -0.12 | -0.24 – (-0.01) | **0.040** |
| Distance from focus | 0.04 | -0.01 – 0.09 | 0.143 |
| At-risk population | -0.10 | -0.23 – 0.03 | 0.135 |
| Working | -0.30 | -0.41 – (-0.20) | **<0.001** |
| Routines | -0.32 | -0.45 – (-0.19) | **<0.001** |
| COVID-19 as dangerous [a lot] | -0.06 | -0.17 – 0.05 | 0.296 |
| COVID-19 as dangerous [slightly or nothing] | -0.16 | -0.31 – (-0.00) | **0.046** |
| **Observations:** 1456 | | | |
| **R² / R² adjusted:** 0.053 / 0.049 | | | |

CI: Confidence interval.

[1]The **intercept** (or constant) is the expected mean value of Y when all X = 0.

The BIS total score is influenced by age (β = -0.12, $p<0.001$), distance from their residence to the petrochemical complex (β = 0.07, $p = 0.016$), having children (β = -0.17, $p = 0.001$), working during lockdown (β = -0.33, $p<0.001$) and establishing routines during that period (β = -0.37, $p<0.001$). All these variables (significant and non-significant) explain almost 10% of the total score variance of the BIS scale ($r^2$ adjusted = 0.096) (see Table 11).

**Hospital Anxiety and Depression Scale (HADS).** The HADS anxiety score is influenced by age (β = 0.21, $p<0.001$), gender (being male) (β = 0.14, $p = 0.013$), living with children (β = 0.17, $p = 0.008$), working during lockdown (β = 0.18, $p = 0.001$), going out to work (β = -0.16, $p = 0.011$), perceiving the COVID-19 situation as very dangerous (β = -0.33, $p<0.001$) and perceiving it as nothing or slightly dangerous (β = 0.31, $p<0.001$). All these variables (significant and non-significant from Table 8) explain nearly 14% of the anxiety score variance ($r^2$ adjusted = 0.137) (see Table 12).

**Table 9. Regression analysis relating to motor impulsivity in the BIS.**

| Predictors | MOTOR IMPULSIVITY | | |
|---|---|---|---|
| | β | CI | p values |
| (Intercept[1]) | 0.70 | 0.39 – 1.02 | **<0.001** |
| Age | -0.07 | -0.13 – (-0.02) | **0.006** |
| Gender [male] | 0.10 | -0.02 – 0.22 | 0.088 |
| Minors in their care | -0.14 | -0.24 – (-0.03) | **0.011** |
| Losing one's job | 0.14 | -0.03 – 0.31 | 0.115 |
| Working | -0.29 | -0.41 – (-0.17) | **<0.001** |
| Sector [no response] | -0.23 | -0.51 – 0.05 | 0.107 |
| Sector [petrochemical complex] | -0.12 | -0.42 – 0.18 | 0.421 |
| Sector [health] | -0.39 | -0.73 – (-0.06) | **0.019** |
| Routines | -0.36 | -0.49 – (-0.23) | **<0.001** |
| **Observations:** 1456 | | | |
| **R² / R² adjusted:** 0.079 / 0.073 | | | |

CI: Confidence interval.

[1]The **intercept** (or constant) is the expected mean value of Y when all X = 0.

**Table 10. Regression analysis relating to cognitive impulsivity in the BIS.**

| Predictors | COGNITIVE IMPULSIVITY | | |
|---|---|---|---|
| | β | CI | p values |
| (Intercept[1]) | 0.40 | 0.26 – 0.54 | **<0.001** |
| Age | -0.24 | -0.29 – (-0.19) | **<0.001** |
| Gender [male] | -0.19 | -0.30 – (-0.07) | **0.001** |
| Distance from petrochemical complex | 0.04 | -0.00 – 0.09 | 0.074 |
| Minors in their care | -0.24 | -0.34 – (-0.14) | **<0.001** |
| Losing one's job | 0.13 | -0.03 – 0.29 | 0.122 |
| Working | -0.27 | -0.41 – (-0.13) | **<0.001** |
| Telecommuting | 0.13 | 0.00 – 0.26 | **0.047** |
| Routines | -0.23 | -0.36 – (-0.11) | **<0.001** |
| Covid-19 as dangerous [a lot] | 0.10 | -0.01 – 0.20 | 0.075 |
| Covid-19 as dangerous [slightly or nothing] | -0.12 | -0.27 – 0.02 | 0.099 |
| **Observations:** 1456 | | | |
| **R$^2$ / R$^2$ adjusted:** 0.147 / 0.141 | | | |

CI: Confidence interval.

[1]The **intercept** (or constant) is the expected mean value of Y when all X = 0.

The HADS depression score is influenced by losing one's job (β = 0.23, $p$ = 0.006) and establishing routines during lockdown (β = 0.28, $p<0.001$). All these variables (significant and non-significant) explain only 2% of the depression score variance ($r^2$ adjusted = 0.021) (see Table 13).

**Emotional Regulation Questionnaire (ERQ).** The expressive suppression score is influenced by gender (being male) (β = 0.48, $p<0.001$), distance from the residence to the petrochemical complex (β = 0.06, $p$ = 0.026), living with people over 65 (β = 0.22, $p$ = 0.007) and telecommuting (β = -0.19, $p<0.001$). These variables (significant and non-significant) explain 6% of the expressive suppression score variance ($r^2$ adjusted = 0.062) (see Table 14).

**Table 11. Regression analysis relating to total impulsivity in the BIS.**

| Predictors | TOTAL IMPULSIVITY | | |
|---|---|---|---|
| | β | CI | p values |
| (Intercept[1]) | 0.62 | 0.48 – 0.77 | **<0.001** |
| Age | -0.12 | -0.17 – (-0.07) | **<0.001** |
| Distance from petrochemical complex | 0.07 | 0.01 – 0.13 | **0.016** |
| Danger perception [slightly or nothing] | -0.09 | -0.22 – 0.03 | 0.150 |
| Danger perception [a lot] | 0.10 | -0.08 – 0.27 | 0.278 |
| Danger perception [no response] | -0.11 | -0.25 – 0.03 | 0.110 |
| Minors in their care | -0.17 | -0.27 – (-0.07) | **0.001** |
| Working | -0.33 | -0.44 – (-0.23) | **<0.001** |
| Routines | -0.37 | -0.50 – (-0.25) | **<0.001** |
| **Observations:** 1456 | | | |
| **R$^2$ / R$^2$ adjusted:** 0.101 / 0.096 | | | |

CI: Confidence interval.

[1]The **intercept** (or constant) is the expected mean value of Y when all X = 0.

**Table 12. Regression analysis relating to anxious symptomatology in the HADS.**

| Predictors | ANXIETY SYMPTOMS | | |
|---|---|---|---|
| | β | CI | p values |
| (Intercept[1]) | -0.08 | -0.21 – 0.04 | 0.195 |
| Age | 0.21 | 0.16 – 0.26 | **<0.001** |
| Gender [male] | 0.14 | 0.03 – 0.25 | **0.013** |
| Distance from petrochemical complex | -0.04 | -0.10 – 0.01 | 0.095 |
| Danger perception [slightly or nothing] | 0.04 | -0.08 – 0.16 | 0.485 |
| Danger perception [a lot] | -0.16 | -0.33 – 0.00 | 0.056 |
| Danger perception [no response] | 0.10 | -0.03 – 0.24 | 0.127 |
| Minors in their care | 0.17 | 0.07 – 0.27 | **0.001** |
| Dependent or disabled people | -0.16 | -0.34 – 0.01 | 0.073 |
| At-risk populations | -0.10 | -0.23 – 0.02 | 0.104 |
| Losing one's job | -0.21 | -0.37 – (-0.05) | **0.008** |
| Working | 0.18 | 0.07 – 0.29 | **0.001** |
| Going out to work | -0.16 | -0.28 – (-0.04) | **0.011** |
| Covid-19 as dangerous [a lot] | -0.33 | -0.43 – (-0.22) | **<0.001** |
| Covid-19 as dangerous [slightly or nothing] | 0.31 | 0.16 – 0.45 | **<0.001** |
| Observations: 1561 | | | |
| R² / R² adjusted: 0.145 / 0.137 | | | |

CI: Confidence interval.

[1]The **intercept** (or constant) is the expected mean value of Y when all X = 0.

The cognitive reappraisal score is influenced by age (β = 0.06, $p$ = 0.029) and by establishing routines during lockdown (β = 0.19, $p$ = 0.006). All the variables entered in the regression analysis (significant and non-significant) explain only 1% of the cognitive reappraisal score ($r^2$ adjusted = 0.014) (see Table 15).

## Discussion

The main aim of this study was to analyse the effects of living near a petrochemical complex on different psychological outcomes (stress, impulsivity, anxiety, depression and emotional regulation strategies) deriving from the COVID-19 pandemic and the consequent lockdown.

**Table 13. Regression analysis relating to depressive symptomatology in the HADS.**

| Predictors | DEPRESSIVE SYMPTOMS | | |
|---|---|---|---|
| | β | CI | p values |
| (Intercept[1]) | -0.33 | -0.47 – (-0.19) | **<0.001** |
| Losing one's job | 0.23 | 0.07 – 0.40 | **0.006** |
| Working | 0.10 | -0.01 – 0.21 | 0.065 |
| Routines | 0.28 | 0.15 – 0.40 | **<0.001** |
| Covid-19 as dangerous [a lot] | 0.11 | -0.00 – 0.21 | 0.057 |
| Covid-19 as dangerous [slightly or nothing] | -0.15 | -0.30 – 0.01 | 0.060 |
| Observations: 1561 | | | |
| R² / R² adjusted: 0.025 / 0.021 | | | |

CI: Confidence interval.

[1]The **intercept** (or constant) is the expected mean value of Y when all X = 0.

**Table 14. Regression analysis relating to suppression strategy in the ERQ.**

| | SUPPRESSION | | |
|---|---|---|---|
| Predictors | β | CI | p values |
| (Intercept[1]) | -0.06 | -0.14 – 0.03 | 0.193 |
| Gender [male] | 0.48 | 0.36 – 0.60 | <**0.001** |
| Distance from petrochemical complex | 0.06 | 0.01 – 0.12 | **0.026** |
| Older than 65 years | 0.22 | 0.06 – 0.39 | **0.007** |
| Dependent or disabled people | 0.17 | -0.04 – 0.38 | 0.107 |
| Telecommuting | -0.19 | -0.29 – (-0.08) | <**0.001** |
| **Observations:** 1347 | | | |
| **R²  / R² adjusted:** 0.065 / 0.062 | | | |

CI: Confidence interval.

[1]The **intercept** (or constant) is the expected mean value of Y when all X = 0.

We hypothesized that subjects living near the complex would show higher levels of psychological impact during lockdown compared to individuals living away from the focus of environmental pollution. The present results indicate that people living closer to petrochemical complexes report greater risk perception. However, in contrast to our hypothesis, no significant relationship between the psychological variables and proximity to the focus was detected when comparing people living near and away from a chemical/petrochemical complex.

Nevertheless, it is worth noting how different conditions during lockdown affected some of the psychological variables studied here. In this regard the data showed that people who were working and going out fairly frequently had a lower perception of stress, whereas those who had lost their jobs had a higher stress perception. Therefore the economic impact of the COVID-19 situation in the area under evaluation could have a deleterious effect on people's health, since stress is broadly related to both physical and psychological diseases (assuming such a differentiation could be made) [46–49]. Moreover, psychological distress is prevalent in frequent users of primary health care and emergency departments and has a significant association with frequent use of these services [50]. This suggests that the increase in distress due to COVID-19 could considerably increase health expenditure.

**Table 15. Regression analysis relating to re-evaluation strategy in the ERQ.**

| | REAPPRAISAL | | |
|---|---|---|---|
| Predictors | β | CI | p values |
| (Intercept[1]) | -0.18 | -0.31 – (-0.04) | **0.009** |
| Age | 0.06 | 0.01 – 0.12 | **0.029** |
| Minors in their care | 0.08 | -0.03 – 0.19 | 0.151 |
| Being sick with Covid-19 | -0.29 | -0.65 – 0.07 | 0.113 |
| At-risk population | -0.11 | -0.25 – 0.03 | 0.135 |
| Going out to work | 0.10 | -0.04 – 0.23 | 0.154 |
| Routines | 0.19 | 0.06 – 0.32 | **0.006** |
| **Observations:** 1347 | | | |
| **R²  / R² adjusted:** 0.018 / 0.014 | | | |

CI: Confidence interval.

[1]The **intercept** (or constant) is the expected mean value of Y when all X = 0.

The results of the present study also reveal that having minors in one's care and establishing routines reduces the total score for impulsivity, together with an increase in planning. This indicates that planning might be an important strategy for avoiding impulsive decisions and actions that are considered risky, maladaptive and symptomatic of various brain disorders such as attention-deficit hyperactivity disorder, drug addiction and affective disorders [51]. A recent study assessing the psychological effects of lockdown in a Spanish population suggested there may be a relationship between the establishment of routines and the participants' levels of resilience, implying a better adaptation to adversity [52].

Losing one's job increased people's total, motor and cognitive impulsivity, indicating that they were trying to find a solution to their family's economic situation. We believe this is an important characteristic, since impulsivity is an important aspect of obsessive-compulsive disorder (OCD) and attention deficit hyperactivity disorder (ADHD) [53–55]. We should also bear in mind that poor executive function increases the likelihood that healthy young adults will engage in risky and potentially dangerous acts [56]. Unfortunately, we cannot avoid the fact that the pandemic will be traumatic for children, and a history of childhood trauma in OCD patients has indirect effects on the severity of the OCD and depressive symptoms and is associated with more severe anxiety, higher levels of impulsivity, a higher prevalence of ADHD and lower levels of education [57].

Our data also show that subjects going out to work decreased both cognitive and total impulsivity, since they had no added economic problems at the time and up to a point still had their old routines. However, it is important to bear in mind that not working or telecommuting increased planning and decreased impulsivity, which at the same time could be a good indicator of mental health.

The data on anxious and depressive symptomatology were very surprising. Our results indicate that losing one's job reduced anxious symptomatology. Cognitive models of social anxiety disorder (SAD) emphasize anticipatory processing as a prominent maintaining factor occurring before social-evaluative events [58]. In addition, as suggested by Wong, McEvoy [59], anticipatory processing reflected by a general repetitive thinking factor had moderately large associations with social anxiety and life interference. Considering all the above, it could be suggested that what increased these symptoms might be worrying about losing one's job, but not the fact of actually losing it.

Another surprising result was that having minors in their care increased people's anxiety symptoms, whereas living with dependent or disabled people decreased them. We believe that this apparently contradictory result may be related to two different things: a) perhaps because minors are more demanding than dependent people, and b) perhaps because dependent subjects are also an at-risk population and, deep down, living with them–when at-risk populations everywhere are falling ill or dying–was actually a relief. Interestingly, we also found that establishing routines during lockdown increased depressive symptomatology. In this regard, although some authors have suggested that positivity is an important attitude in resilient people [60] and others have suggested there is a relationship between resilience levels and the establishment of routines [52], the increase in depressive symptomatology should not be interpreted negatively. Since our results are not related to the presence of depressive syndrome but only to depressive symptomatology, the results might indicate an increase in awareness of the difficult times we are living through, suggesting also that a realistic perspective in life is associated with resilience.

Emotional regulation strategies were also affected by some differential conditions. Living with people over 65 years old and dependent people increased suppression, while establishing routines increased re-evaluation strategies. People living with dependent and/or old people tend to suppress emotions as a regulation strategy. One explanation may be that they do not

want to show their real emotions in front of their relatives (old/dependent people). However, establishing routines, which increases planning, could make it possible to re-evaluate the difficult situation that we were experiencing and focus more on the benefits rather than the problems. It is important to consider that avoidance strategies increase anticipatory anxiety [61], which could be an important factor in the development and maintenance of anxiety disorders.

As for the regression models implemented, we cannot ignore the fact that the variables used in the present study explain only 22% of the variance in the PSS, 10% in the BIS, and just 2% in the HADS and 1% in the ERQ. These results mean that the items used in this survey may not be the most appropriate for detecting changes in the psychological variables due to the lockdown situation generated by COVID-19. Moreover, statistical analysis showed no significant relationship between the risk perception of COVID-19 and any other variables included in the Petrocovid Survey.

In conclusion, the results of the present study have indicated that living near a big petrochemical complex did not have any additional adverse psychological influence on the general population in Catalonia during the first lockdown due to COVID-19. Although we can conclude that the lockdown conditions included in this survey were mainly related to changes in the impulsivity levels of the participants, we would also suggest that the economic effects are going to be harder than those initially detected in this study. However, some limitations should be considered when interpreting our results. The first limitation concerns the sampling method. Due to the restrictions imposed during lockdown, the current study was performed using a non-probabilistic sampling method which limits the generalization of our results to the general population. Second, the sample of the study consists mainly of women. Thus the results are not representative of the male population. Finally, the high intensity of the emotional impact deriving from the measures adopted to control the COVID outbreak could have masked the psychological effects related to living close to petrochemical complexes. To overcome these limitations and corroborate our results, more studies using probabilistic sampling methods conducted in post-lockdown periods are necessary. Nevertheless, despite the limitations, the present study points to the presence of psychological effects produced by the lockdown procedures and notes the need to conduct follow-up studies to better understand the psychological impact of these measures. This knowledge could be useful to generate prevention strategies for mental health and to minimize the impact of the COVID outbreak on the general population's well-being.

## Acknowledgments

The authors would like to thank Toni Masip and Montse Marquès for their technical support in the development and dissemination of the Petrocovid Survey (PS).

## Author Contributions

**Conceptualization:** Paloma Vicens, Luis Heredia, José L. Domingo, Margarita Torrente.

**Data curation:** Paloma Vicens, Luis Heredia, Edgar Bustamante, Yolanda Pérez, Margarita Torrente.

**Formal analysis:** Paloma Vicens, Luis Heredia, Edgar Bustamante, Yolanda Pérez, Margarita Torrente.

**Funding acquisition:** José L. Domingo.

**Methodology:** Paloma Vicens, Luis Heredia, Margarita Torrente.

**Writing – original draft:** Paloma Vicens, Luis Heredia.

**Writing – review & editing:** Paloma Vicens, Luis Heredia, Yolanda Pérez, José L. Domingo, Margarita Torrente.

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
