## [Decision Letter · Decision Letter 0]

19 Jan 2021

PONE-D-20-36639

Does living close to petrochemical complex increase the adverse psychological effects of COVID-19 lockdown?

PLOS ONE

Dear Dr.Torrente,

Thank you for submitting your manuscript to PLOS ONE. After careful consideration, we feel that it has merit but does not fully meet PLOS ONE’s publication criteria as it currently stands. Therefore, we invite you to submit a revised version of the manuscript that addresses the points raised during the review process.

We look forward to receiving your revised manuscript.

Kind regards,

Flávia L. Osório, PhD

Academic Editor

PLOS ONE

Additional Editor Comments:

The reviewers considered the article suitable for publication, but suggested revising small topics, in particular, a review of the writing.

2.) We suggest you thoroughly copyedit your manuscript for language usage, spelling, and grammar. If you do not know anyone who can help you do this, you may wish to consider employing a professional scientific editing service.  

3.) Please provide additional details regarding participant consent. In the ethics statement in the Methods and online submission information, please ensure that you have specified what type you obtained (for instance, written or verbal, and if verbal, how it was documented and witnessed). If your study included minors, state whether you obtained consent from parents or guardians. If the need for consent was waived by the ethics committee, please include this information.

4.) We note that Figure 2 in your submission contains map images which may be copyrighted. All PLOS content is published under the Creative Commons Attribution License (CC BY 4.0), which means that the manuscript, images, and Supporting Information files will be freely available online, and any third party is permitted to access, download, copy, distribute, and use these materials in any way, even commercially, with proper attribution. For these reasons, we cannot publish previously copyrighted maps or satellite images created using proprietary data, such as Google software (Google Maps, Street View, and Earth). For more information, see our copyright guidelines: http://journals.plos.org/plosone/s/licenses-and-copyright.

a.) You may seek permission from the original copyright holder of Figure 2 to publish the content specifically under the CC BY 4.0 license. 

b.) If you are unable to obtain permission from the original copyright holder to publish these figures under the CC BY 4.0 license or if the copyright holder’s requirements are incompatible with the CC BY 4.0 license, please either i) remove the figure or ii) supply a replacement figure that complies with the CC BY 4.0 license. Please check copyright information on all replacement figures and update the figure caption with source information. If applicable, please specify in the figure caption text when a figure is similar but not identical to the original image and is therefore for illustrative purposes only.

Reviewers' comments:

Reviewer's Responses to Questions

**Comments to the Author**

1. Is the manuscript technically sound, and do the data support the conclusions?

Reviewer #1: Partly

Reviewer #2: Yes

2. Has the statistical analysis been performed appropriately and rigorously? 

Reviewer #1: Yes

Reviewer #2: Yes

3. Have the authors made all data underlying the findings in their manuscript fully available?

Reviewer #1: No

Reviewer #2: Yes

4. Is the manuscript presented in an intelligible fashion and written in standard English?

Reviewer #1: No

Reviewer #2: Yes

5. Review Comments to the Author

Reviewer #1: Thank you for the opportunity to review your manuscript.

In this paper, the authors examined differential effects on different psychological outcomes derived from the COVID-19 pandemic and the consequent lockdown in individuals living close to petrochemical complex and subjects living in other locations in Catalonia, Spain. They found that greater risk perception was reported by those living closer to petrochemical complexes. However, the relationships between psychological outcomes and living near to chemical/petrochemical complex were not significant.

My comments:

The abstract lacks any numerical data.

The primary problem with this manuscript is that there are grammar errors and word flow issues. Unfortunately, these distract from the paper's quality and make certain portions challenging to understand what authors are trying to convey. The manuscript needs thorough editing to address a number of

grammatical errors and to improve readability.

Reviewer #2: Overall, the manuscript is well written, and the community-based research is need of the day. However, few of the suggestions are:

a) the authors need to add more community-based studies to strengthen the importance of study framework.

b) There is a difference between objective and hypothesis. The authors have overlapped the two as highlighted in line 588 “We hypothesized”, but I am unable to find any hypothesizing throughout the article. Make a single or two clear objectives and then test different hypothesis of the study accordingly instead of making objectives 1,2,3….. which is not right.

c) Limitations & Future directions should be added

d) Add Implications of the study along with recommendations

e) Enhance the conclusion

f) Manuscript needs some revamping of sentences at places

6. PLOS authors have the option to publish the peer review history of their article (what does this mean?). If published, this will include your full peer review and any attached files.

Reviewer #1: No

Reviewer #2: No

---

## [Author Response · Author response to Decision Letter 0]

22 Feb 2021

RESPONSE TO REVIEWERS

The manuscript has been accurately revised following PLOS ONE’s style requirements. Tables are now embedded as Microsoft tables.

2.) We suggest you thoroughly copyedit your manuscript for language usage, spelling, and grammar. If you do not know anyone who can help you do this, you may wish to consider employing a professional scientific editing service. 

“Servei Lingüístic” (Language Service) from Universitat Rovira i Virgili

http://www.llengues.urv.cat/en/about/servei-linguistic/

A copy with highlighted changes is included (corrections in green, from Peter of the URV Language Service), uploaded as a supporting information file.

A clean copy with the changes is submitted (uploaded as the new manuscript file).

3.) Please provide additional details regarding participant consent. In the ethics statement in the Methods and online submission information, please ensure that you have specified what type you obtained (for instance, written or verbal, and if verbal, how it was documented and witnessed). If your study included minors, state whether you obtained consent from parents or guardians. If the need for consent was waived by the ethics committee, please include this information.

This study did not include minors. The participant consent was obtained via the own online survey. If the participant did not give the consent or he/she was minor, the survey finished with no data from them being recorded.

In the revised version of the manuscript, we have included -Material and Methods section- the following statement: “If the participants were minors or did not accept the ethics consent on the online platform, the survey ended without any data being collected. All the data obtained came from participants over the age of 18 who consented to their data being collected by the researchers.” 

4.) We note that Figure 2 in your submission contains map images which may be copyrighted. All PLOS content is published under the Creative Commons Attribution License (CC BY 4.0), which means that the manuscript, images, and Supporting Information files will be freely available online, and any third party is permitted to access, download, copy, distribute, and use these materials in any way, even commercially, with proper attribution. For these reasons, we cannot publish previously copyrighted maps or satellite images created using proprietary data, such as Google software (Google Maps, Street View, and Earth). For more information, see our copyright guidelines: http://journals.plos.org/plosone/s/licenses-and-copyright.

We appreciate your comment and all the suggestions and information. 

The base map used in figure 2 is the “Municipal, provincial and autonomous limits” from the Instituto Geográfico Nacional of Spain. It is possible to use that map under a creative common license. 

Products: https://www.ign.es/resources/licencia/Condiciones_licenciaUso_IGN.pdf, 

License of use: https://www.ign.es/resources/licencia/Condiciones_licenciaUso_IGN.pdf

We have uploaded the permission document from the Spanish Ministerio de Fomento, where the use under CC BY 4.0 license is assured.

Also, the following explanation has been added to Figure 2. 

Fig 2. Distance between the place of residence (survey locations) and the nearest source of petrochemical pollutants. Map based on BDLJE CC-BY scne.es 

Reviewers' comments:

Reviewer's Responses to Questions

Comments to the Author

1. Is the manuscript technically sound, and do the data support the conclusions?

Reviewer #1: Partly

Reviewer #2: Yes

We expect that after the corrections/changes done, Reviewer #1 will find the manuscript more complete in this sense.

2. Has the statistical analysis been performed appropriately and rigorously? 

Reviewer #1: Yes

Reviewer #2: Yes

--

3. Have the authors made all data underlying the findings in their manuscript fully available?

Reviewer #1: No

Reviewer #2: Yes

All the information to replicate the study was already available. In order to follow all the recommendations, we decided to upload the metadata file with the raw data collected from the survey in the DAS repository https://dans.knaw.nl/nl
https://easy.dans.knaw.nl/ui/mydatasets

under the following identification number: 

https://doi.org/10.17026/dans-za8-qpmw

4. Is the manuscript presented in an intelligible fashion and written in standard English?

Reviewer #1: No

Reviewer #2: Yes

According to the Editor and Reviewers’ suggestions/comments we have employed a Language Service in order to revise all the manuscript for English grammar and typographical errors, as well as to assure a clear English in the text.

5. Review Comments to the Author 

Reviewer #1: Thank you for the opportunity to review your manuscript.

In this paper, the authors examined differential effects on different psychological outcomes derived from the COVID-19 pandemic and the consequent lockdown in individuals living close to petrochemical complex and subjects living in other locations in Catalonia, Spain. They found that greater risk perception was reported by those living closer to petrochemical complexes. However, the relationships between psychological outcomes and living near to chemical/petrochemical complex were not significant.

Comments:

The abstract lacks any numerical data.

We have revised the abstract and included some numerical data from the results section.

The primary problem with this manuscript is that there are grammar errors and word flow issues. Unfortunately, these distract from the paper's quality and make certain portions challenging to understand what authors are trying to convey. The manuscript needs thorough editing to address a number of

grammatical errors and to improve readability.

Following Editor and Reviewers’ suggestions/comments, we have now employed a Language Service in order to revise all the manuscript for English grammar, typographical errors and to assure a clear English in the revised manuscript. 

Reviewer #2: Overall, the manuscript is well written, and the community-based research is need of the day. However, few of the suggestions are:

a) the authors need to add more community-based studies to strengthen the importance of study framework.

The authors have included more community-based studies in the introduction section. However, it is important to consider that nowadays, there is a lack of studies focused on the same topic.

“However, the working memory function is not the only neuropsychological function that can be affected by the presence of air pollutants in the area surrounding petrochemical complexes. In a community-based previous study by Vichit-Vadakan, Vajanapoom [15], a sample of 17,515 participants living within a 10-km radius of petrochemical industries were assessed for neuropsychological performance. The results showed that those living near these complexes performed worse on tests that assessed eye-hand coordination, short-term recall, and hand and eye movement responsiveness. Moreover, participants living less than 3 km from the centre of the industrial complex were more likely to exhibit forgetfulness, anxiety, depression and loss of concentration. Finally, it is important also to consider the possible impact on the population’s mental health stemming from the occurrence of accidents in nearby petrochemical complexes. A community-based study by Peek, Cutchin [16] reported decreases in local residents’ self-perceived mental and physical health after experiencing an incident at the plant, even in those who were not directly affected by it.”

“These studies could be pointing to a serious effect on the population’s depressive mood due to the lockdown procedures during the COVID-19 outbreak. In this regard, a recent review of community-based studies on the prevalence of depression during 2020 has reported a prevalence of 25%, a percentage seven times higher than the global estimate for depression in 2017 (3.44%) [26].”

b) There is a difference between objective and hypothesis. The authors have overlapped the two as highlighted in line 588 “We hypothesized”, but I am unable to find any hypothesizing throughout the article. Make a single or two clear objectives and then test different hypothesis of the study accordingly instead of making objectives 1,2,3….. which is not right.

The revised manuscript differentiates between objectives (main and specific objectives) and hypothesis (tested by the statistics tests). 

c) Limitations & Future directions should be added 

The following limitations have been added to the manuscript: “However, some limitations should be considered when interpreting our results. The first limitation concerns the sampling method. Due to the restrictions imposed during lockdown, the current study was performed using a non-probabilistic sampling method which limits the generalization of our results to the general population. Second, the sample of the study consists mainly of women. Thus the results are not representative of the male population. Finally, the high intensity of the emotional impact deriving from the measures adopted to control the COVID outbreak could have masked the psychological effects related to living close to petrochemical complexes. To overcome these limitations and corroborate our results, more studies using probabilistic sampling methods conducted in post-lockdown periods are necessary.”

d) Add Implications of the study along with recommendations 

The authors include in the manuscript the following text: “Nevertheless, despite the limitations, the present study points to the presence of psychological effects produced by the lockdown procedures and notes the need to conduct follow-up studies to better understand the psychological impact of these measures. This knowledge could be useful to generate prevention strategies for mental health and to minimize the impact of the COVID outbreak on the general population’s well-being”.

e) Enhance the conclusion 

Limitations and Implications have been added to the conclusion, completing it.

f) Manuscript needs some revamping of sentences at places

The authors have carefully revised the text of manuscript, which has been also done by the “Servei Lingüístic” (Language Service) from Universitat Rovira i Virgili. This has been done in order to improve all the poor sentences and expressions.

---

## [Decision Letter · Decision Letter 1]

11 Mar 2021

Does living close to a petrochemical complex increase the adverse psychological effects of the COVID-19 lockdown?

PONE-D-20-36639R1

Dear Dr. Torrente,

We’re pleased to inform you that your manuscript has been judged scientifically suitable for publication and will be formally accepted for publication once it meets all outstanding technical requirements.

Kind regards,

Flávia L. Osório, PhD

Academic Editor

PLOS ONE

Additional Editor Comments (optional):

Reviewers' comments:

Reviewer's Responses to Questions

**Comments to the Author**

1. If the authors have adequately addressed your comments raised in a previous round of review and you feel that this manuscript is now acceptable for publication, you may indicate that here to bypass the “Comments to the Author” section, enter your conflict of interest statement in the “Confidential to Editor” section, and submit your "Accept" recommendation.

Reviewer #1: All comments have been addressed

Reviewer #2: All comments have been addressed

2. Is the manuscript technically sound, and do the data support the conclusions?

Reviewer #1: Yes

Reviewer #2: Yes

3. Has the statistical analysis been performed appropriately and rigorously? 

Reviewer #1: Yes

Reviewer #2: Yes

4. Have the authors made all data underlying the findings in their manuscript fully available?

Reviewer #1: No

Reviewer #2: Yes

5. Is the manuscript presented in an intelligible fashion and written in standard English?

Reviewer #1: Yes

Reviewer #2: Yes

6. Review Comments to the Author

Reviewer #1: (No Response)

Reviewer #2: The manuscript titled "Does living close to a petrochemical complex increase the adverse psychological effects of the COVID-19 lockdown?" is in a acceptable form to be published.

7. PLOS authors have the option to publish the peer review history of their article (what does this mean?). If published, this will include your full peer review and any attached files.

Reviewer #1: No

Reviewer #2: No

---

## [Editor Report · Acceptance letter]

15 Mar 2021

PONE-D-20-36639R1 

Does living close to a petrochemical complex increase the adverse psychological effects of the COVID-19 lockdown? 

Dear Dr. Torrente:

I'm pleased to inform you that your manuscript has been deemed suitable for publication in PLOS ONE. Congratulations! Your manuscript is now with our production department. 

Kind regards, 

on behalf of

Dr. Flávia L. Osório 

Academic Editor

PLOS ONE